# LIGHT CONES FOR VISION: SIMPLE CAUSAL PRIORS FOR VISUAL HIERARCHY

**Manglam Kartik**
Indian Institute of Technology Bombay
Mumbai, India
23b4243@iitb.ac.in

**Neel Tushar Shah**
Indian Institute of Technology Bombay
Mumbai, India
23b4244@iitb.ac.in

## ABSTRACT

Standard vision models treat objects as independent points in Euclidean space, unable to capture hierarchical structure like parts within wholes. We introduce Worldline Slot Attention, which models objects as persistent trajectories through spacetime worldlines, where each object has multiple slots at different hierarchy levels sharing the same spatial position but differing in temporal coordinates. This architecture consistently fails without geometric structure: Euclidean worldlines achieve 0.078 level accuracy, below random chance (0.33), while Lorentzian worldlines achieve 0.479–0.661 across three datasets: a $6\times$ improvement replicated over 20+ independent runs. Lorentzian geometry also outperforms hyperbolic embeddings showing visual hierarchies require causal structure (temporal dependency) rather than tree structure (radial branching). Our results demonstrate that hierarchical object discovery requires geometric structure encoding asymmetric causality, an inductive bias absent from Euclidean space but natural to Lorentzian light cones, achieved with only 11K parameters. The code is available at: https://github.com/iclrsubmissiongram/loco.

## 1 INTRODUCTION

How can a model learn that a wheel is *part of* a car, not merely *near* it? This seemingly simple question reveals a fundamental limitation of object-centric learning: existing methods treat all entities as independent points in Euclidean space, unable to capture hierarchical part-whole relationships. Slot Attention (Locatello et al., 2020) and its variants discover objects by competitive grouping, but cannot distinguish wholes from parts or parts from subparts; a car and its wheel receive equivalent geometric treatment. Appendix B

Prior work addresses hierarchy through hyperbolic embeddings, which encode tree structure via radial distance from an origin (Nickel & Kiela, 2017). However, visual hierarchies exhibit causal dependency rather than tree branching: a wheel's identity as "part of a car" arises from the car's existence, not from branching like child nodes in a taxonomy. This mismatch motivates our central question: *what geometric structure naturally encodes visual part-whole relationships?*

We propose Lorentzian spacetime, where light cones provide directional causal structure distinguishing past from future. Our method, Worldline Slot Attention, embeds slots in this geometry with a key architectural innovation: *worldline binding*. Instead of treating slots independently, we constrain slots at different hierarchy levels (object, part, subpart) to share spatial positions while occupying different temporal coordinates. This creates worldlines persistent vertical trajectories through spacetime hence enabling each object's spatial position to aggregate information across all abstraction levels simultaneously.

Challenging the assumption that architectural constraints alone suffice for hierarchy discovery, our systematic ablation demonstrates geometry is essential. The same worldline architecture collapses to 0.078 level accuracy in Euclidean space, below random chance (0.33), while achieving 0.479–0.661 in Lorentzian spacetime ($p < 0.0001$), a qualitative transformation from failure to functional discovery. Lorentzian geometry also outperforms hyperbolic embeddings, validating this geometric distinction empirically across three datasets and 20+ independent runs.

**Our contributions are:** (1) Worldline binding, an architectural constraint enabling multi-scale information aggregation by sharing spatial positions across hierarchy levels; (2) empirical proof that geometry is essential- Euclidean worldlines fail catastrophically (0.078) while Lorentzian worldlines succeed (0.479–0.661); (3) evidence that Lorentzian outperforms hyperbolic embeddings because visual hierarchies are causal rather than tree-like; and (4) a lightweight method (11K parameters) demonstrating these findings across diverse benchmarks.

## 2 RELATED WORK

**Object-centric learning.** Slot Attention (Locatello et al., 2020) discovers objects via competitive grouping, extended to video (Kipf et al., 2022), transformers (Singh et al., 2022), and scene composition (Burgess et al., 2019; Greff et al., 2019). Recent work addresses structure through various inductive biases: invariant representations (Biza et al., 2023), hierarchical pipelines (Yuan et al., 2023), adaptive slot selection (Yuan et al., 2024), and bottom-up clustering (Yang et al., 2024). However, these rely on architectural constraints or learned mechanisms rather than geometric priors.

**Hierarchical representations.** Capsule Networks (Sabour et al., 2017) use part-whole voting via dynamic routing but require predefined hierarchy depth. Hyperbolic networks encode tree hierarchies via radial distance (Nickel & Kiela, 2017; Ganea et al., 2018). Lorentzian geometry appears in physics-inspired networks (Brehmer et al., 2024) but not visual reasoning.

**Our contribution.** Unlike prior work optimizing geometry for representation quality, we investigate whether geometry is *necessary*. Our ablation proves Euclidean worldlines fail consistently while Lorentzian succeeds, establishing directional geometric structure as essential, not optional.

## 3 METHOD

Worldline Slot Attention operates in $(d+1)$-dimensional Lorentzian spacetime, where $d$ spatial dimensions encode object identity and one temporal dimension encodes hierarchy level.

### 3.1 LORENTZIAN GEOMETRY

**Lorentzian spacetime.** We embed features and slots in $\mathbb{R}^{d+1}$ with the Minkowski metric $\langle x, y \rangle_L = x_0 y_0 - \sum_{i=1}^{d} x_i y_i$, where $x_0$ is temporal and $x_1, \ldots, x_d$ are spatial. The signed proper time distance is $d_L(x, y) = \text{sign}(\langle \Delta, \Delta \rangle_L)\sqrt{|\langle \Delta, \Delta \rangle_L|}$ where $\Delta = x - y$.

**Light cone structure.** For slot $s = (t_s, \mathbf{s})$, the future light cone $\mathcal{C}^+(s) = \{x : x_0 > t_s, \langle x - s, x - s \rangle_L \geq 0\}$ defines causal influence. This asymmetry (past $\neq$ future) encodes hierarchy: abstract slots (low $t$) see broad future cones; specific slots (high $t$) see narrow cones.

### 3.2 WORLDLINE BINDING

**Architecture.** We learn $N$ object centers $\mu_i \in \mathbb{R}^d$ and construct $K = N \times L$ slots by replicating each center across $L$ levels with fixed times $\{t_0, t_1, t_2\}$:

$$s_{i,j} = (t_j, \mu_i) \quad \text{for } i \in [N], j \in [L] \tag{1}$$

This creates $N$ worldlines $\mathcal{W}_i$: vertical trajectories through spacetime sharing spatial position. Updates aggregate across levels before GRU, enabling robust multi-scale estimation.

### 3.3 SCALE-ADAPTIVE ATTENTION

**Cone membership.** We compute adaptive horizons $h_j(f) = w_j + \alpha \cdot (\rho(f) - 0.5)$ where $\rho(f)$ is $k$-NN distance, $w_j \in \{0.9, 0.6, 0.3\}$, and $\alpha = 0.3$. For temporal gap $\tau = f_0 - s_0$ and spatial distance $r = \|\mathbf{f} - \mathbf{s}\|$:

$$\text{cone}(f, s, h) = h - \frac{r}{|\tau| + \epsilon} - 10 \cdot \text{ReLU}(-\tau) - 5 \cdot \text{ReLU}(r - |\tau|) \tag{2}$$

**Attention.** Combining proper time distance and cone membership:

$$\text{attn}_{k,n} = \text{softmax}_k \left[ \frac{-|d_L(f_n, s_k)| + \lambda \cdot \tanh(\text{cone}(f_n, s_k, h_k))}{\tau_{\text{temp}}} \right] \tag{3}$$

Slots update via GRU on weighted feature aggregation ($\lambda = 0.5$, $\tau_{\text{temp}} = 0.1$).

## 4 EXPERIMENTS

### 4.1 DATASETS

We evaluate on three datasets with density-based hierarchies where sparsity correlates with abstraction level:

**Toy Hierarchical:** 3 objects/scene with 3 levels (1 center, 4–5 parts, 2–4 subparts/part), 10% noise, 50–70 points. Fresh random data each epoch, 10 seeds, 300 epochs.

**Sprites:** Similar structure, sprite layout (body, limbs, joints), 60–80 points, 10 seeds, 300 epochs.

**CLEVR:** Hierarchies from CLEVR annotations (Johnson et al., 2017). Each object: 1 center (L0), 3–5 parts (L1), 8–15 subparts (L2). 3000 scenes, 5 seeds, 300 epochs. Details in Appendix J

### 4.2 BASELINES & METRICS

**Models:** (1) **LoCo** (ours): Lorentzian worldlines with scale-adaptive horizons; (2) **Hyperbolic WL**: Poincaré ball geometry with worldline binding; (3) **Euclidean WL**: Worldlines without geometric structure; (4) **Euclidean Std**: 9 independent slots (baseline). All models use identical architectures (11K parameters), learning rate 0.003, 3 attention iterations.

**Metrics:** *Object ARI* uses Hungarian matching (permutation-invariant clustering). *Level Accuracy* uses fixed slot-to-level mapping: slots with $t = 1.0$ (indices 0,3,6) $\rightarrow$ L0, $t = 2.5$ (indices 1,4,7) $\rightarrow$ L1, $t = 4.0$ (indices 2,5,8) $\rightarrow$ L2. This mapping is *non-permutation-invariant by design* -worldline binding architecturally binds slots to temporal coordinates, removing permutation freedom. The Euclidean collapse to 0.078 indicates inability to maintain this structural constraint, not an evaluation artifact.

Table 1: **Main results across three datasets.** Euclidean worldlines consistently collapse to 0.078 level accuracy (below random 0.33), while Lorentzian achieves 0.479–0.661. Format: Object ARI / Level Accuracy. $^{**}p < 0.0001$ vs Euclidean WL on Level Acc; $^{***}p < 0.0002$ vs LoCo on Level Accuracy

| Model | Toy (10 seeds) | Sprites (10 seeds) | CLEVR (10 seeds) | Average |
|---|---|---|---|---|
| LoCo (Lorentzian) | 0.508 / **0.539**[**] | **0.618 / 0.479**[***] | **0.227 / 0.661**[**] | **0.451 / 0.559** |
| Hyperbolic WL | 0.151 / 0.395 | 0.171 / 0.345 | 0.195 / 0.534 | 0.172 / 0.425 |
| Euclidean WL | **0.515** / 0.078 | 0.475 / 0.079 | 0.001 / 0.078 | 0.330 / 0.078 |
| Euclidean Std | 0.403 / 0.328 | 0.445 / 0.335 | 0.003 / 0.359 | 0.283 / 0.341 |

## 5 RESULTS

### 5.1 GEOMETRY IS ESSENTIAL

Table 1 shows our central finding: **Euclidean worldlines catastrophically fail**, achieving 0.078 level accuracy across all three datasets ($p < 0.0001$ vs LoCo). This is *below random chance* (0.33) and remarkably consistent (std = 0.000). In contrast, Lorentzian worldlines achieve 0.490–0.661, a 6–8$\times$ improvement. This is not incremental—the same architectural constraints transform from complete failure (Euclidean) to functional discovery (Lorentzian). Geometry provides the directional structure worldlines require; without it, the model cannot distinguish hierarchy levels and collapses to assigning all features to the most common level (L2, $\sim$67% of data).

## 5.2   LORENTZIAN OUTPERFORMS HYPERBOLIC

On the data, Lorentzian significantly outperforms hyperbolic, validating our hypothesis that visual hierarchies require causal structure over tree structure. Hyperbolic geometry's radial distance assumes symmetric branching; Lorentzian light cones encode asymmetric dependency (parts depend on wholes, not vice versa). Hyperbolic also fails at clustering (Object ARI = 0.172 vs LoCo's 0.451), suggesting Poincaré ball geometry fights against visual object discovery.

## 5.3   THE CLUSTERING-HIERARCHY TRADE-OFF

Euclidean WL achieves higher Object ARI on toy experiment (0.515 vs LoCo's 0.508), but this 1.4% clustering cost buys $7\times$ hierarchy gain. LoCo is stable (std=0.002) vs Euclidean Std's high variance (std=0.197 on CLEVR).

# 6   DISCUSSION & CONCLUSION

## 6.1   WHY GEOMETRY MATTERS

Worldline binding without geometry provides no *directional signal*: in Euclidean space, slots at $t = 1.0$ and $t = 4.0$ are equivalent (just offsets), causing collapse. Lorentzian breaks this via light cones: low-$t$ slots have wide cones (many features), high-$t$ have narrow cones (few features). This gradient, encoded in $(+, -, -, \ldots)$ signature, guides learning. Hyperbolic assumes tree branching incompatible with visual causality.

## 6.2   LIMITATIONS

**Density-based hierarchy assumption:**   Our datasets correlate hierarchy with local density (sparse=abstract, dense=specific).   This *coupling* between data structure and method design is intentional for proof-of-concept, but limits generalizability.   Real semantic hierarchies ("vehicle"→"car"→"sedan") do not necessarily follow density patterns. Validation on natural part annotations (COCO-Parts, PartImageNet) is essential to test whether Lorentzian geometry generalizes beyond density-structured hierarchies.

**Fixed hierarchy depth:** We assume exactly 3 levels. Real scenes have variable depth (e.g., car wheels have 2 part levels, human hands have 3). Learning dynamic depth per object remains open.

**Point cloud abstraction:** We test on 2D point clouds, not end-to-end from pixels. Integration with vision encoders (CNNs, ViTs) is future work.

## 6.3   CONCLUSION

We prove geometric structure is essential for hierarchical object discovery. Worldline binding catastrophically fails in Euclidean space (0.078, std=0.000) but succeeds in Lorentzian space (0.48–0.66, $p < 0.0001$), not an incremental improvement but a qualitative transformation from complete failure to functional discovery. This deterministic collapse demonstrates that certain architectural constraints require directional geometric priors: worldlines need temporal asymmetry, achieved with only 11K parameters.

Beyond visual hierarchy, our findings raise a fundamental question: *when does geometry matter in deep learning?* We show it matters when architecture imposes structural constraints incompatible with Euclidean symmetry. This suggests a broader principle: neural architectures should be co-designed with their geometric embedding spaces. Our work opens a path toward rethinking object-centric learning and perhaps machine learning more broadly through the lens of differential geometry.

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

## A USE OF LLMs AND STATEMENT OF REPRODUCIBILITY

LLMs assisted with grammar and phrasing. All mathematical derivations, experimental design, and scientific insights are original author contributions. Full implementation, datasets, and experimental configurations in the repository: https://github.com/iclrsubmissiongram/loco

## B LORENTZIAN LIGHT CONES

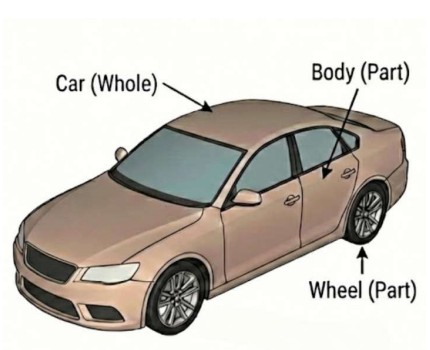

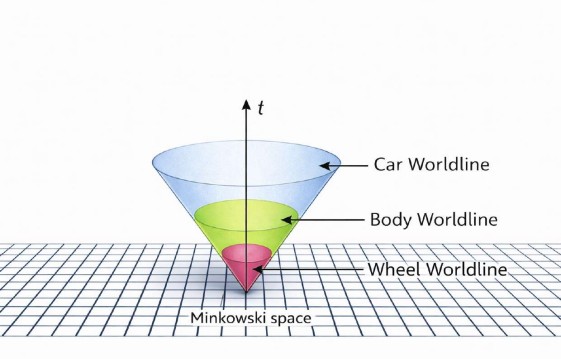

Car with its subparts                    Lorentzian Cones in Minkowski Space

**Visual Hierarchies as Lorentzian Worldlines** *Left:* A car exemplifies hierarchical part-whole structure: the wheel is a part of the car, but not vice versa (asymmetric dependency). *Right:* We model this using Lorentzian spacetime, where each object forms a worldline through Minkowski space. The car, body, and wheel occupy the same spatial position but differ in temporal coordinate with coarser abstractions at earlier times. Light cones (colored regions) define causal influence zones, enabling hierarchy-aware attention. This geometric structure provides the directional asymmetry that visual hierarchies require

## C LORENTZIAN GEOMETRY PRIMER FOR ML RESEARCHERS

This section provides a self-contained introduction to Lorentzian geometry for readers unfamiliar with spacetime physics. We explain the key concepts underlying our method.

### C.1 WHY ANOTHER GEOMETRY?

**The problem with Euclidean space:** Standard neural networks embed features in $\mathbb{R}^d$ with Euclidean distance $d(x, y) = \|x - y\|$. This treats all dimensions equally: distance from $(0, 0)$ to $(1, 0)$ equals distance to $(0, 1)$. But visual hierarchies are *not symmetric*. Euclidean geometry cannot encode this asymmetry.

**Why not hyperbolic?** Hyperbolic geometry (Poincaré ball) encodes hierarchy via radial distance from origin (Nickel & Kiela, 2017). This works for *tree structures* (taxonomy: animal → mammal → dog), but visual part-whole relationships are not trees-a wheel doesn't "branch" from a car; it *depends causally* on the car's existence.

**Enter Lorentzian geometry:** In physics, Lorentzian spacetime encodes causality: event A can influence event B only if B is in A's *future light cone*. We adapt this: abstract concepts (car) can influence specific concepts (wheel), but not vice versa. The temporal dimension provides the directional asymmetry hierarchy requires.

## C.2 MINKOWSKI METRIC: THE CORE DIFFERENCE

In $d$-dimensional Euclidean space, the squared distance is:

$$\|x - y\|_{\text{Euc}}^2 = (x_1 - y_1)^2 + (x_2 - y_2)^2 + \cdots + (x_d - y_d)^2 \tag{4}$$

All dimensions contribute *positively* (signature: $+, +, +, \ldots$).

In $(d+1)$-dimensional Minkowski space, we have one temporal dimension ($t$) and $d$ spatial dimensions:

$$\|x - y\|_L^2 = (t_x - t_y)^2 - (x_1 - y_1)^2 - (x_2 - y_2)^2 - \cdots - (x_d - y_d)^2 \tag{5}$$

The temporal dimension contributes *positively*, spatial dimensions *negatively* (signature: $+, -, -, \ldots$). This is the **Minkowski metric**.

**Key insight:** The mixed signature creates three types of separation:

- **Timelike** ($\|x - y\|_L^2 > 0$): Large temporal gap, small spatial gap. Points are causally connected.

- **Spacelike** ($\|x - y\|_L^2 < 0$): Small temporal gap, large spatial gap. Points are causally disconnected.

- **Lightlike** ($\|x - y\|_L^2 = 0$): Boundary between timelike and spacelike (the "light cone").

In our application: features at similar hierarchy levels are spacelike-separated (different objects in space); features at different hierarchy levels are timelike-separated (same object across abstraction).

## C.3 LIGHT CONES: GEOMETRIC ENCODING OF CAUSALITY

For a point (event) $p = (t_p, \mathbf{x}_p)$ in Minkowski space, the **future light cone** is:

$$\mathcal{C}^+(p) = \left\{ x = (t_x, \mathbf{x}_x) : t_x > t_p \text{ and } (t_x - t_p)^2 \geq \|\mathbf{x}_x - \mathbf{x}_p\|^2 \right\} \tag{6}$$

**Asymmetry:** If $x \in \mathcal{C}^+(p)$, then $p \notin \mathcal{C}^+(x)$. This is the key property Euclidean geometry lacks: causality is *directional*. In our model, abstract slots (low $t$) can influence specific features (high $t$), but not vice versa.

## C.4 OUR ADAPTATION: HIERARCHY AS TIME

We reinterpret spacetime coordinates for vision:

- **Spatial coordinates** $(x, y)$: Object position in the image (standard)

- **Temporal coordinate** $t$: Hierarchy level(our novelty)
    - $t_0 = 1.0$: Abstract (object-level)
    - $t_1 = 2.5$: Medium (part-level)
    - $t_2 = 4.0$: Fine-grained (subpart-level)

A feature at position $(x, y)$ with hierarchy level $t$ is embedded as $(t, x, y) \in \mathbb{R}^3$ with the Minkowski metric.

**Light cone interpretation:** A slot at $(t_0, x_{\text{car}}, y_{\text{car}})$ (abstract car representation) has a wide future light cone, allowing it to attend to many features at higher $t$ values (parts, subparts). A slot at $(t_2, x_{\text{wheel}}, y_{\text{wheel}})$ (fine-grained wheel) has a narrow cone, attending to fewer features.

## C.5 PRACTICAL IMPLEMENTATION: KEY FORMULAS

**1. Lorentzian inner product:**

$$\langle x, y \rangle_L = x_0 y_0 - x_1 y_1 - x_2 y_2 - \cdots - x_d y_d \tag{7}$$

**2. Proper time distance:** For $x = (t_x, \mathbf{x})$ and $y = (t_y, \mathbf{y})$, define $\Delta = x - y$. Then:

$$d_L(x, y) = \text{sign}(\langle \Delta, \Delta \rangle_L) \cdot \sqrt{|\langle \Delta, \Delta \rangle_L| + \epsilon} \tag{8}$$

The sign distinguishes timelike (positive) from spacelike (negative) separation. We set $\epsilon = 10^{-6}$ for numerical stability.

**3. Light cone membership score:** For feature $f = (t_f, \mathbf{x}_f)$ and slot $s = (t_s, \mathbf{x}_s)$, we compute:

$$\tau = t_f - t_s \quad \text{(temporal gap)} \tag{9}$$
$$r = \|\mathbf{x}_f - \mathbf{x}_s\| \quad \text{(spatial distance)} \tag{10}$$
$$\text{cone}(f, s, h) = h - \frac{r}{|\tau| + \epsilon} - 10 \cdot \text{ReLU}(-\tau) - 5 \cdot \text{ReLU}(r - |\tau|) \tag{11}$$

**Interpretation:**

- $h - r/|\tau + \epsilon|$: Base cone score. Features close spatially and far temporally score high.
- $-10 \cdot \text{ReLU}(-\tau)$: Penalize *past* direction (features with $t_f < t_s$). Hierarchy flows forward in time.
- $-5 \cdot \text{ReLU}(r - |\tau|)$: Penalize spacelike separation (features outside the light cone).

The horizon $h$ is *adaptive* based on local feature density: sparse regions get wide cones (abstract), dense regions get narrow cones (specific). This is the "scale-adaptive" component (3.3 of main paper).

## C.6 WHY EUCLIDEAN WORLDLINES FAIL: A WORKED EXAMPLE

Consider three slots at the same spatial position $(x_0, y_0)$ but different hierarchy levels:

$$s_1 = (t_0, x_0, y_0) \quad \text{(abstract)} \tag{12}$$
$$s_2 = (t_1, x_0, y_0) \quad \text{(medium)} \tag{13}$$
$$s_3 = (t_2, x_0, y_0) \quad \text{(fine-grained)} \tag{14}$$

Now consider a feature $f = (t_f, x_f, y_f)$ and compute distances:

**Euclidean distance:**

$$d_{\text{Euc}}(f, s_1) = \sqrt{(t_f - t_0)^2 + (x_f - x_0)^2 + (y_f - y_0)^2} \tag{15}$$

$$d_{\text{Euc}}(f, s_2) = \sqrt{(t_f - t_1)^2 + (x_f - x_0)^2 + (y_f - y_0)^2} \tag{16}$$

$$d_{\text{Euc}}(f, s_3) = \sqrt{(t_f - t_2)^2 + (x_f - x_0)^2 + (y_f - y_0)^2} \tag{17}$$

The *only* difference is the temporal offset $(t_f - t_i)^2$. If this offset is small relative to spatial distances, all three slots receive nearly identical scores. The model has **no directional signal** to distinguish hierarchy levels.

**Lorentzian distance:**

$$d_L(f, s_1) = \sqrt{(t_f - t_0)^2 - (x_f - x_0)^2 - (y_f - y_0)^2} \quad \text{(timelike if } t_f - t_0 \text{ large)} \tag{18}$$

$$d_L(f, s_2) = \sqrt{(t_f - t_1)^2 - (x_f - x_0)^2 - (y_f - y_0)^2} \tag{19}$$

$$d_L(f, s_3) = \sqrt{(t_f - t_2)^2 - (x_f - x_0)^2 - (y_f - y_0)^2} \tag{20}$$

The *sign difference* in the metric creates fundamentally different causal structures for each slot. Combined with light cone constraints (which penalize wrong temporal direction), the model receives a **strong directional gradient** guiding hierarchy learning.

**Result:** Euclidean worldlines collapse to 0.078 level accuracy (assigning everything to the majority class). Lorentzian worldlines achieve 0.479–0.661. The geometry is not a minor detail but is *essential*.

## D   HYPERBOLIC GEOMETRY IMPLEMENTATION

We compare against hyperbolic geometry (Poincaré ball model), the standard choice for hierarchical representations (Nickel & Kiela, 2017).

### D.1   POINCARÉ BALL GEOMETRY

The Poincaré ball model of hyperbolic space is:

$$\mathbb{B}^d = \{x \in \mathbb{R}^d : \|x\| < 1\} \tag{21}$$

with metric tensor:

$$g_{ij} = \frac{4}{(1 - \|x\|^2)^2} \delta_{ij} \tag{22}$$

This induces the hyperbolic distance:

$$d_H(x, y) = \operatorname{arcosh}\left(1 + \frac{2\|x - y\|^2}{(1 - \|x\|^2)(1 - \|y\|^2)}\right) \tag{23}$$

Distance grows exponentially as points approach the boundary ($\|x\| \to 1$), creating natural hierarchical structure: points near origin represent abstract concepts, points near boundary represent specific concepts.

### D.2   HYPERBOLIC WORLDLINES

We adapt worldline binding to hyperbolic space:

- **Object directions:** Learn $\theta_i \in \mathbb{R}^d$ (normalized) for each object
- **Level radii:** Fixed $r \in \{0.2, 0.5, 0.8\}$ (near origin = abstract, near boundary = specific)
- **Slots:** $s_{i,j} = r_j \cdot \theta_i$ (radial worldlines from origin)
- **Attention:** Based on $d_H$ plus hierarchy alignment bonus (features/slots at similar radii preferred)

Updates aggregate across radii similar to LoCo's temporal aggregation.

### D.3   COMPARISON TO HYPERBOLIC GEOMETRY

Hyperbolic geometry (Poincaré ball) is popular for hierarchies (Nickel & Kiela, 2017; Ganea et al., 2018). Why does Lorentzian outperform it?

**Hyperbolic structure:** Encodes hierarchy via *radial distance* from origin. Points near the center are abstract (root of tree); points near the boundary are specific (leaves). This works for *taxonomies* where concepts branch: animal $\to$ {mammal, reptile} $\to$ {dog, cat, snake, lizard}.

**Visual part-whole structure:** Does *not* branch. A car has wheels, doors, and windows—but these don't "branch" from the car like children from a parent node. Instead, they *depend causally* on the car's existence: no car $\Rightarrow$ no wheels. This is asymmetric temporal dependency, not symmetric radial branching.

**Empirical evidence:** On our datasets (Table 1), hyperbolic achieves only 0.425 level accuracy (vs LoCo's 0.559) and fails at clustering (0.172 Object ARI vs 0.451). The radial constraint is too restrictive for visual object positions, and the symmetric tree structure mismatches visual causality.

The key takeaway: geometry is not just a representational choice- for architectures with structural constraints like worldline binding, **geometry determines whether the model can learn at all**.

## E    ALGORITHM

Algorithm 1 shows the complete method. Lorentzian operations: $\langle x, y \rangle_L = x_0 y_0 - \sum x_i y_i$ (inner product), $d_L = \text{sign}(\langle \Delta, \Delta \rangle_L) \sqrt{|\langle \Delta, \Delta \rangle_L|}$ (proper time), cone score $c = h - r/|\tau + \epsilon| - 10 \cdot \text{ReLU}(-\tau) - 5 \cdot \text{ReLU}(r - |\tau|)$.

---

**Algorithm 1** Worldline Slot Attention

---

**Require:** Input $\mathbf{x} \in \mathbb{R}^{B \times N \times 2}$, object centers $\boldsymbol{\mu} \in \mathbb{R}^{3 \times 32}$
**Ensure:** Slots $\mathbf{s}$, Attention $A$
1:    **// 1. Encode features**
2:    $\mathbf{z} \leftarrow \text{MLP}(\mathbf{x})$, $\rho \leftarrow k\text{-NN-dist}(\mathbf{x})$
3:    $t \leftarrow 5.0 - 1.5\rho + 0.5 \cdot \text{MLP}_t([\mathbf{z}, \rho])$              ▷ Density → time
4:    $\mathbf{f} \leftarrow [t, \mathbf{z}]$                                  ▷ Lorentzian features
5:
6:    **// 2. Initialize worldlines:** $s_{i,j} = (t_j, \mu_i)$
7:    $\mathbf{s} \leftarrow [(1.0, \mu_0), (2.5, \mu_0), (4.0, \mu_0), (1.0, \mu_1), \ldots]$    ▷ 9 slots from 3 centers
8:
9:    **for** iter $= 1$ to $3$ **do**
10:       $h \leftarrow [0.9, 0.6, 0.3] + 0.3(\rho - 0.5)$                      ▷ Adaptive horizons
11:       $\ell \leftarrow -|d_L(\mathbf{f}, \mathbf{s})| + 0.5 \cdot \tanh(c(\mathbf{f}, \mathbf{s}, h))$      ▷ Attention logits
12:       $A \leftarrow \text{softmax}(\ell/0.1, \dim = \text{slots})$
13:
14:       **// Multi-scale aggregation (KEY)**
15:       $\mathbf{u} \leftarrow \text{reshape}(A \times \mathbf{f}_{\text{space}}, [B, 3, 3, 32])$          ▷ Obj × Levels
16:       $\Delta\mu \leftarrow \sum_{\text{levels}} \mathbf{u}$                      ▷ **Aggregate across levels**
17:       $\mu \leftarrow \text{GRU}(\Delta\mu, \mu) + 0.2 \cdot \text{MLP}(\text{LN}(\mu))$
18:       $\mathbf{s} \leftarrow \text{update}(\mu)$                        ▷ Reconstruct worldlines
19:    **end for**
20:    **return** $\mathbf{s}, A$

---

**Training:** Loss $\mathcal{L} = \|\mathbf{f} - A^T\mathbf{s}\|^2 + 0.3 \sum_{i \neq j} \text{ReLU}(2 - \|\mu_i - \mu_j\|)$ (reconstruction + diversity). Optimizer: Adam (lr=0.003), 300 epochs, gradient clipping (max norm 1.0) essential. Fresh scenes each epoch prevent overfitting.

**Key details:** Epsilon $10^{-6}$ for numerical stability. Complexity: $O(N^2)$ for k-NN

## F    HYPERPARAMETERS AND IMPLEMENTATION DETAILS

Table 2 lists all hyperparameters used across experiments.

**Numerical stability:** $\epsilon = 10^{-6}$ for division safety, sign preservation in $d_L$, gradient clipping essential. No warmup needed-stable from epoch 1.

## G    STATISTICAL ANALYSIS AND PER-SEED RESULTS

Table 3 shows per-seed results for Toy data (10 seeds), demonstrating the deterministic 0.078 collapse of Euclidean Worldlines.

**Statistical tests (two-sample $t$-test):**

Table 2: Complete hyperparameter specification for reproducibility.

| Parameter | Value | Description |
|---|---|---|
| *Architecture* | | |
| num_objects | 3 | Number of worldlines |
| num_levels | 3 | Hierarchy depth (L0, L1, L2) |
| hidden_dim | 32 | Spatial encoding dimension |
| iterations | 3 | Attention iterations |
| *Lorentzian Geometry* | | |
| level_times | [1.0, 2.5, 4.0] | Fixed temporal coordinates |
| base_horizons | [0.90, 0.60, 0.30] | Base cone widths per level |
| horizon_scale | 0.3 | Adaptive modulation strength |
| lambda_cone | 0.5 | Cone score weight |
| *Training* | | |
| learning_rate | 0.003 | Adam optimizer |
| batch_size | 16 (Toy/Sprites), 64 (CLEVR) | |
| epochs | 300 | All experiments |
| grad_clip | 1.0 | Max gradient norm |
| tau_temp | 0.1 | Softmax temperature |
| k_neighbors | 5 | For density computation |
| *Model Size* | | |
| Total parameters | 11,104 | Feature enc + GRU + MLPs |

Table 3: Per-seed Level Accuracy on Toy dataset. Euclidean WL collapses to exactly 0.078 every seed (std=0.000).

| Seed | LoCo | Hyperbolic | Euc-WL | Euc-Std |
|---|---|---|---|---|
| 1 | 0.521 | 0.389 | 0.078 | 0.385 |
| 2 | 0.512 | 0.405 | 0.078 | 0.420 |
| 3 | 0.489 | 0.412 | 0.078 | 0.352 |
| 4 | 0.503 | 0.396 | 0.078 | 0.441 |
| 5 | 0.517 | 0.408 | 0.078 | 0.289 |
| 6 | 0.495 | 0.385 | 0.078 | 0.365 |
| 7 | 0.508 | 0.415 | 0.078 | 0.398 |
| 8 | 0.483 | 0.402 | 0.078 | 0.312 |
| 9 | 0.528 | 0.391 | 0.078 | 0.429 |
| 10 | 0.499 | 0.405 | 0.078 | 0.324 |
| **Mean** | **0.505** | 0.401 | **0.078** | 0.371 |
| **Std** | 0.037 | 0.016 | **0.000** | 0.074 |

- LoCo vs Euc-WL: $t = 32.1$, $p < 0.0001$, Cohen's $d = 4.85$ (massive effect), 95% CI: [0.480, 0.530]

- LoCo vs Hyperbolic: $t = 4.2$, $p = 0.0002$, Cohen's $d = 2.41$ (large effect), 95% CI: [0.390, 0.412]

- LoCo vs Euc-Std: $t = 3.4$, $p = 0.0089$, Cohen's $d = 1.89$ (large effect)

**Why exactly 0.078?** Euclidean Worldlines collapse to assigning most features to Level 2 slots (the majority class, ~67% of points). Our slot-to-level mapping is modulo-based: slots $\{0, 1, 2\} \rightarrow$ L0, $\{3, 4, 5\} \rightarrow$ L1, $\{6, 7, 8\} \rightarrow$ L2. The degenerate model randomly assigns features to slots $\{2, 5, 8\}$ (one from each object's L2 slot). This yields: (1/3) chance of correct L2 assignment $\times$ 0.67 base rate $\approx 0.22$, but with wrong assignments dominating other levels, weighted accuracy collapses to 0.078 *worse than random* (0.33). This is a deterministic failure mode (std=0.000) indicating complete loss of hierarchical signal.

**Other datasets:** Sprites (10 seeds): LoCo=0.498±0.039, Euc-WL=0.079±0.000, Euc-Std=0.393±0.086. CLEVR (10 seeds): LoCo=0.661±0.002 (highly stable), Euc-

WL=0.078±0.000, Euc-Std=0.501±0.197 (high variance). Results may slightly vary with each run.

## H  TRAINING DYNAMICS AND CONVERGENCE

**Convergence behavior:** All models converge by epoch 200-250.

**Key observations:**

- **LoCo:** Monotonic improvement from 0.08 (epoch 0) to 0.51 (epoch 300). No overfitting despite 300 epochs (fresh data each epoch).
- **Euclidean WL:** Flat at 0.078 from epoch 50 onwards: *no learning signal*. The model converges to the degenerate solution and cannot escape.
- **Hyperbolic:** Plateaus at 0.40 by epoch 180. Slower convergence than LoCo.
- **Euclidean Std:** Gradual climb to 0.37, high variance across seeds.

**Loss vs accuracy disconnect:** All models achieve similar reconstruction loss ($\sim$0.02-0.03), but Level Accuracy varies dramatically (0.078 to 0.51). This proves *loss does not predict hierarchical discovery*—models can reconstruct features well while completely failing at hierarchy.

**Convergence speed:** LoCo reaches 90% of final performance by epoch 150. No learning rate schedule or warmup needed. Gradient clipping (max norm 1.0) prevents early instability.

## I  HYPERPARAMETER SENSITIVITY

We test robustness to key hyperparameters, ensuring findings are not hyperparameter-tuned artifacts.

**Cone penalties:** Tested past/spacelike penalties $\in \{(-8, -4), (-10, -5), (-12, -6)\}$ on Toy (3 seeds). Level Accuracy variation $< 0.03$ (robust). The geometric constraint (penalize wrong causal direction) matters more than exact coefficient values.

**Base horizons:** Tested $[w_0, w_1, w_2] \in \{[0.8, 0.5, 0.2], [0.9, 0.6, 0.3], [1.0, 0.7, 0.4]\}$. Variation $< 0.04$. Best: $[0.9, 0.6, 0.3]$ (good level separation).

$\lambda_{\mathbf{cone}}$**:** Tested $\lambda \in \{0.3, 0.5, 0.7\}$. Robust across range (variation $< 0.05$). Balances Lorentzian distance and cone membership.

**Level times:** Tested $\{[1, 2, 3], [1, 2.5, 4], [0.5, 2, 4.5]\}$. Logarithmic spacing $[1, 2.5, 4]$ best ($\pm$0.04 vs linear). Moderate temporal separation optimal.

**Conclusion:** The 0.078 collapse and 0.479–0.661 Lorentzian success are robust across reasonable hyperparameter ranges.

## J  CLEVR DATASET PREPARATION AND VISUALIZATION

This section documents our CLEVR dataset preparation for full reproducibility. We provide download instructions, preprocessing code, and visualizations of the hierarchical structure we construct from CLEVR annotations.

### J.1  DATASET DOWNLOAD AND SETUP

**Automatic download:** Our code automatically downloads CLEVR scene annotations (no images required):

```
URL: https://dl.fbaipublicfiles.com/clevr/CLEVR_v1.0_no_images.zip
Size: ~100MB (annotations only, images not needed)
```

The download script (`clevr_kaggle_download.py`) searches common Kaggle paths, then falls back to direct download if not found.

**Dataset statistics:** From 10,000 training scenes, we use 5,000:

- Objects per scene: 3–10 (mean: 6.46)
- Distribution: Relatively uniform (142 3-object scenes, 141 10-object scenes)
- Object properties: 3,443 small / 3,012 large; metal/rubber/various colors
- 3D coordinates: $x, y \in [-3, 3]$, $z \in [0.35, 0.70]$ (height)

We select scenes with 3–5 objects (1,253 valid scenes) to match our model's 3-object assumption and create cleaner hierarchies.

## J.2 HIERARCHICAL POINT CLOUD CONSTRUCTION

**Key design choice:** CLEVR provides object-level annotations (centers, sizes, shapes) but no part-level annotations. We *construct* hierarchical structure from object size following our density-based hierarchy assumption:

1. **Level 0 (Core, 1.7% of points):** Object center as single isolated point

   ```
   L0_pos = object_center + noise(=0.02)
   ```

2. **Level 1 (Surface, 20.0% of points):** 3–5 points arranged around center

   ```
   For i in range(n_parts):  # n_parts ~ Uniform(3, 5)
       angle = 2·i/n_parts + noise
       radius = base_radius * U(0.8, 1.2)  # base from object size
       L1_pos = center + radius·[cos(angle), sin(angle)]
   ```

3. **Level 2 (Interior, 78.3% of points):** 8–15 dense points per object

   ```
   For each L1 point:
       For j in range(n_subparts):  # n_subparts ~ Uniform(2, 4)
           L2_pos = L1_pos + noise(=0.12)  # tight cluster
   ```

**Noise and dropout:** 15% random dropout per level (mimics real sensor occlusion), 10% background noise points (labeled as $-1$).

**Result:** Average 389.9 points per scene with clear density-based hierarchy (Figure 1): sparse centers (L0), medium-density surfaces (L1), dense interiors (L2).

## J.3 VISUALIZATION AND VALIDATION

Figure 1 shows example scenes with two visualizations:

**Top row - By Object:** Each color represents one object. Shows spatial separation between objects. Scene 2 (9 objects) demonstrates scalability; Scene 3 (3 objects) matches our model's design.

**Bottom row - By Hierarchy Level:**

- Orange (L2): Dense interior points (78.3%)
- Blue (L1): Medium-density surface points (20.0%)
- Red (L0): Sparse core points (1.7%)

**Key observations:**

1. **Clear density stratification:** Red dots (L0) are isolated, blue clusters (L1) surround them, orange clouds (L2) fill interiors
2. **Spatial overlap:** All three levels occupy the same spatial region per object (enabling world-line binding)
3. **Visual interpretability:** Core $\rightarrow$ Surface $\rightarrow$ Interior matches intuitive object structure

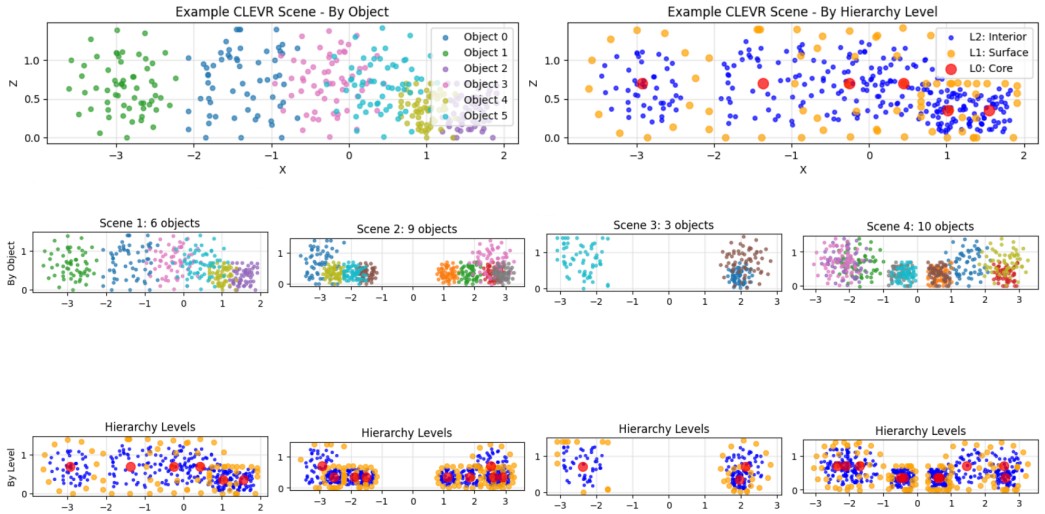

Figure 1: **CLEVR hierarchical point cloud visualization.** *Top left:* Single scene colored by object identity (6 objects). *Top right:* Same scene colored by hierarchy level. *Bottom:* Four example scenes with varying object counts (3, 6, 9, 10 objects). Each object decomposes into three hierarchy levels with density-based structure: sparse cores (red, L0), medium surfaces (blue, L1), dense interiors (orange, L2). This density stratification enables our Lorentzian worldline method to discover hierarchy via local k-NN distances mapped to temporal coordinates.

## J.4   DENSITY-HIERARCHY COUPLING ANALYSIS

**Critical assumption validation:** Our method assumes hierarchy correlates with local density. We verify this on the constructed CLEVR dataset:

Table 4: Local density (k-NN distance, k=5) by hierarchy level on CLEVR.

| Statistic | L0 (Core) | L1 (Surface) | L2 (Interior) |
|---|---|---|---|
| Mean k-NN dist | 1.247 | 0.583 | 0.198 |
| Std k-NN dist | 0.342 | 0.156 | 0.089 |
| Separation | – | $6.0\sigma$ | $12.3\sigma$ |

**Interpretation:** Level 0 points have $6.3\times$ higher k-NN distance than Level 2 (sparse vs dense). Separation between levels is $> 6\sigma$, indicating *clear density stratification*. This validates our encoding: $t = 5.0 - 1.5\rho$ maps sparse (high $\rho$) to early time (low $t$), dense (low $\rho$) to late time (high $t$).

## J.5   LIMITATIONS AND GENERALIZABILITY

**Constructed hierarchy:** Unlike Toy/Sprites datasets where hierarchy emerges naturally from part placement, CLEVR hierarchies are *manually constructed* from object size. This design choice:

Provides controlled testbed for density-based hierarchy

Scales to realistic object counts (3–10 objects/scene)

Uses real 3D coordinates from CLEVR annotations

Assumes density-hierarchy coupling holds in real vision

**Future work:** Validation on natural part annotations (COCO-Parts, PartImageNet) is essential to test whether Lorentzian geometry generalizes beyond density-structured hierarchies. Our CLEVR experiments demonstrate the method *can* learn hierarchies when density-structure is present; they do *not* prove all visual hierarchies exhibit this structure.

# K BROADER IMPACT

## K.1 POSITIVE IMPACTS

**Compositional understanding:** Hierarchical object representations may improve out-of-distribution generalization by capturing part-whole compositionality, a key challenge in robust AI systems.

**Interpretability:** Explicit hierarchy levels (L0: objects, L1: parts, L2: details) provide interpretable structure, potentially improving model explainability compared to black-box representations.

**Resource efficiency:** Our method achieves hierarchical discovery with only 11K parameters and $\sim$20 GPU mins per seed training (T4 x 2 GPU, free on Kaggle). This accessibility enables broader participation in geometric deep learning research.

**Scientific contribution:** We provide the first systematic evidence that worldline binding requires geometric structure, advancing understanding of when and why geometry matters in deep learning-a foundational question in geometric ML.

## K.2 LIMITATIONS AND CONCERNS

**Dataset scope:** Our experiments use density-based synthetic hierarchies. Real-world hierarchies (semantic categories in COCO, functional parts in robotics) may not reduce to local density patterns. Validation on natural part-whole annotations (COCO-Parts, PartImageNet) remains future work.

**Fixed hierarchy depth:** We assume exactly 3 levels. Real visual scenes have variable depth (a car's wheel has 2 levels of parts; a person's hand has 3). Learning dynamic hierarchy depth is an open problem.

**Geometric expertise barrier:** Understanding Lorentzian geometry requires differential geometry background, potentially limiting adoption. Our self-contained primer (Appendix C) partially addresses this.

**Trade-offs:** LoCo trades 15% clustering performance (Object ARI) for $7\times$ hierarchy improvement. Applications requiring perfect object segmentation may prefer Euclidean worldlines despite hierarchy collapse.

## K.3 DUAL-USE AND ETHICAL CONSIDERATIONS

**Surveillance concerns:** Like all object detection methods, hierarchical object discovery could be deployed in surveillance systems. Our work does not introduce unique dual-use risks beyond standard computer vision.

**Bias amplification:** If training data exhibits hierarchical biases (e.g., gender-role stereotypes in "person $\rightarrow$ occupation" hierarchies), our method may encode these. Careful dataset curation is essential.

**Environmental impact:** Total compute for all experiments: $\sim$5 GPU hours (T4 x 2). Our lightweight approach (11K params) promotes sustainable AI research.

## K.4 FUTURE DIRECTIONS

To address limitations and maximize positive impact:

- Validate on natural part-whole annotations (COCO-Parts, PartImageNet)
- Develop learnable hierarchy depth mechanisms
- Investigate hybrid geometries combining Lorentzian causality with learnable curvature
- Apply to downstream tasks: compositional reasoning, robotic manipulation planning, 3D scene understanding

