# OpenReview forum: "LIGHT CONES FOR VISION: SIMPLE CAUSAL PRIORS FOR VISUAL HIERARCHY"
_ICLR.cc/2026/Workshop/GRaM — ICLR 2026 Workshop GRaM Poster_

### Official Review · Reviewer_uJqS · 2026-02-16
**Clean evidence that Lorentzian/light-cone geometry is the key inductive bias for LoCo in this setup, but the hierarchy signal may be entangled with the density-->time construction---decoupling that and adding robust metrics/visuals would strengthen the claim.**

**Rating:** 7
**Confidence:** 3

**Review:**

Summary. This paper proposes Worldline Slot Attention (“LoCo”), an object-centric architecture that binds slots across hierarchy levels by sharing spatial coordinates while separating them along a temporal axis, and argues that Lorentzian (light-cone) geometry provides the essential asymmetric/causal inductive bias needed for part–whole hierarchy discovery. Across three point-cloud hierarchy datasets (Toy, Sprites, CLEVR-derived), the key empirical claim is striking: the same “worldline” architecture fails deterministically in Euclidean space (level accuracy ≈ 0.078, below chance), while Lorentzian embeddings succeed (≈ 0.48–0.66), and also outperform a hyperbolic alternative.

Strengths: (1)  Clear, falsifiable hypothesis + strong ablation: the “Euclidean worldlines collapse vs Lorentzian succeed” result is unusually crisp and well-documented (incl. per-seed stats). (2) Minimal, lightweight setup (≈11K params) that isolates the geometric prior rather than relying on scale. (3) Helpful primer and conceptual figure connecting part–whole asymmetry to light-cone causality (appendix figure).
Main concerns / limitations: (1) Hierarchy is largely “density-coded.” The method explicitly maps local density (kNN distance) into “time,” so it’s not yet clear how much of the gain is causal geometry vs density-to-level supervision by construction. The paper acknowledges this, but it is central to the claims. (2) Metric choice may bias interpretation. “Level accuracy” uses a fixed slot-to-level mapping (non-permutation-invariant). While this is motivated by worldline binding, it would help to show that conclusions hold under a more permutation-robust hierarchical metric. (3) Not end-to-end from pixels, and hierarchy depth is fixed (3 levels), limiting immediate relevance to mainstream vision pipelines.

Suggested additional experiments:  We would like to suggest some additional experiments to see if the value holds and to further understand the model: (1) Decouple density from hierarchy (stress test): Construct a dataset where levels have matched local density (e.g., resample/perturb points so L0/L1/L2 have similar kNN distributions) and check whether Lorentzian still helps. (2) Learnable vs fixed times: Replace fixed with learned level times (or learned monotone times per worldline) to test whether geometry still matters when “time” is not hard-coded. (3) Permutation-invariant hierarchy evaluation. Report a Hungarian-matched level metric or a hierarchical clustering score (e.g., level-wise ARI with best assignment, or a tree-consistency metric), alongside the current fixed mapping. (4) Scaling and mismatch robustness. Evaluate when the assumed number of objects/levels is wrong (e.g., 3-slot worldlines on scenes with 4–6 objects; variable part counts), to assess practicality. (5) A small proof-of-concept on COCO-Parts / PartImageNet-style annotations (even if only keypoint/part subsets) would strongly support the “visual hierarchies are causal” narrative. The paper already flags this as essential future work.

Suggested visualizations: Qualitative decompositions per dataset: show point clouds colored by predicted object and predicted level (similar to the CLEVR construction plot, but using model outputs). Also,  heatmaps of cone-membership scores or attention weights across levels for representative scenes; include a Euclidean vs Lorentzian side-by-side to illustrate why collapse happens. Lastly, an histogram of inferred feature times t(and their correlation with kNN density) per dataset; this clarifies how much “time = density” drives success would be insightful.

Overall: a neat, well-motivated geometric ablation with unusually clean results, but the current evidence primarily supports “Lorentzian geometry solves density-structured hierarchy discovery under worldline binding.” Strengthening claims about general visual part---whole hierarchy likely requires at least one density-decoupled stress test and a small real-annotation pilot.

**Pmlr Suitability:**

NA

---

### Official Review · Reviewer_79Mz · 2026-02-22
**A nice paper with some caveats to watch out for.**

**Rating:** 6
**Confidence:** 4

**Review:**

**Strengths:**

The authors take a reasonable hypothesis and argue for Lorentzian spacetime being an appropriate geometry to capture hierarchical dependencies of part-whole relantionships.

Experimentally, the setup that evaluates the paper's main claim, including data sets and choice of baselines, is sensible.

I also appreciate that the authors include a very thoughtful list of limitations that readers might be interested in. In particular, I appreciate the discussion on the density and hierarchy.

**Weaknesses:**

The paper is overall well-written, except for the method description. While the Appendix provides the necessary details for a good understanding of the method, what was left in Section 3 is quite sparse and difficult to read. I believe a higher-level description with more cohesive beginning, middle, and end, deferring details to Appendix, would be a more productive use of the space. In particular, it is unclear that the experiments were conducted on point clouds until the very end of the paper, in limitations.


**Questions:**

1. How are the point clouds created from image data?
2. Why is the hyperbolic so much worse than the Euclidean baselines? Is this a problem in the setup itself that would, for instance, change significantly for larger models or settings with different hierarchical levels?

**Minor comments:**

- l. 125: Sentence is missing a period

**Pmlr Suitability:**

NA

---

### Meta-Review · Area_Chair_YzJo · 2026-02-28

**Decision:**

Accept

**Metareview:**

The paper is well written and suitable for gram workshop. I recommend an accept.

**Relevance To Proceedings:**

Tiny paper — does not apply

**Relevance To Workshop:**

Yes — suitable for GRaM

---

### Decision · Program_Chairs · 2026-03-02

Accept (Poster)